# RNA-Dependent RNA Targeting by CRISPR-Cas Systems: Characterizations and Applications

**DOI:** 10.3390/ijms24086894

**Published:** 2023-04-07

**Authors:** Natalia Gunitseva, Marta Evteeva, Anna Borisova, Maxim Patrushev, Fedor Subach

**Affiliations:** Complex of NBICS Technologies, National Research Center “Kurchatov Institute”, 123182 Moscow, Russiapatrushev_mv@nrcki.ru (M.P.)

**Keywords:** CRISPR/Cas-based detection, RNA-targeting systems, biosensors, Cas proteins, diagnostics, *trans*-cleavage

## Abstract

Genome editing technologies that are currently available and described have a fundamental impact on the development of molecular biology and medicine, industrial and agricultural biotechnology and other fields. However, genome editing based on detection and manipulation of the targeted RNA is a promising alternative to control the gene expression at the spatiotemporal transcriptomic level without complete elimination. The innovative CRISPR-Cas RNA-targeting systems changed the conception of biosensing systems and also allowed the RNA effectors to be used in various applications; for example, genomic editing, effective virus diagnostic tools, biomarkers, transcription regulations. In this review, we discussed the current state-of-the-art of specific CRISPR-Cas systems known to bind and cleave RNA substrates and summarized potential applications of the versatile RNA-targeting systems.

## 1. Introduction

CRISPR-Cas system is the protection of natural populations of most bacteria and archaea to foreign nucleic acids [1]. Cas proteins provide microorganisms with RNA-guided adaptive immunity by directing nucleases to bind and cut specific nucleic acid sequences [2]. The feature of this system is the presence of a CRISPR array and adjacent cas genes that form one or more operons. The CRISPR arrays consist of direct repeats separated by spacers (Figure 1), enabling the cells to efficiently detect and destroy earlier encountered pathogens [3]. In general, the genome editing technology based on the CRISPR-Cas system includes two obligate components, single guide RNA (sgRNA) and an effector (a protein or protein complex) that cleavages the target nucleic acid. Genome editing technologies that are currently available and described have a fundamental impact on the development of molecular biology and medicine, industrial and agricultural biotechnology and other fields. Today, tools based on the CRISPR-Cas system are widely used in experimental and applied research [4]. Usually, DNA molecules play the role of the substrate in the reaction as in case of the effectors of type II (Cas9) and V (Cas12a) proteins [5,6]. DNA cleavage activities of CRISPR–Cas systems ensured rapid adoption for genome engineering applications across biology. However, genome editing based on the detection and manipulation of the targeted RNA is a promising alternative to control the gene expression at the spatiotemporal transcriptomic level [7]. Moreover, RNA-editing enzymes do not carry the risk of permanent off-target changes in the genome and are ideal for situations in which homology-directed repair of DNA is not realizable, such as in terminally differentiated (neurons and muscle cells) [8]. Additionally, targeted messenger-RNA edits could be used to recover functions of genes that are impossible to replace by gene therapy [9,10,11,12,13]. Thus far, only Cas proteins of type III [14,15], V [10,16], VI [17] and some orthologs of Cas9 type II [18,19,20] were considered to have RNase activity. 

## 2. Applications of the Versatile CRISPR-Cas RNA Targeting Systems

Unlike DNA editing, RNA editing offers an alternative to genome editing in certain applications in an invertible and controllable manner [21] (Figure 2). First of all, recognition of RNA molecules circumvents hindrance by DNA modifications; for example, chromatin accessibility [22]. Secondly, RNA-targeting effectors are PAM-independent generally, making the chance of runaway mutant variants less likely [23]. That is why the use of RNA-editing technologies is growing rapidly nowadays and may provide many capabilities. CRISPR-Cas RNA-targeting systems allow targeting of nucleic acid fragments specifically including RNA molecules. It permits cleaving RNA in response to finding a target. Such activity also can be modified through development of CRISPR-Cas RNA-targeting systems with dead nuclease activity (dCas). These two properties exactly condition the whole vastness of applications.

### 2.1. Cas-Mediated Biosensing

Biosensing techniques provide the industrial importance to control biomolecules involved in physiological and pathological processes in living organisms, holding great value in basic biochemical research, drug discovery, pollutant detection and monitoring of disease and treatment. The surprising discovery of collateral nucleic acid cleavage by type V and type VI effectors [24,25] enzymes could initially be considered as a flaw. Nevertheless, the ability of some nucleases to exhibit collateral activity could be used as specific and adaptable biosensing tools. Cas-mediated biosensing is based on the catalytic ability of Cas effectors to degrade both target and collateral RNA with single nucleotide sensitivity. Using non-selective cleavage of ssRNA upon target recognition, researchers can determine the amount of any given nucleic acid species in the test sample from fluorescence or colourimetric lateral flow readout.

#### 2.1.1. Bactericidal Agents

The problem of the appearance of antibiotic-resistant pathogenic bacteria leads to the need to develop new methods to tackle them. The properties of the CRISPR-Cas system allow the possibility to adapt it to antibacterial protection. The first attempts to adapt CRISPR-Cas systems to target antimicrobial resistance (AMR) genes led to different outcomes including cytotoxicity and even cell death induction. It became possible to overcome these problems by creating an RNA-effector system. In contrast to DNA-targeting, the RNA targeting is not dependent on PAM sequences. Additionally, the reason why Cas13-based antimicrobial agents received widespread use is the target molecule. Since RNA and not DNA is targeted, the threat of developing resistance due to CRISPR-Cas-induced mutations in DNA is reduced. Finally, the obtained experimental data suggest that Cas13-based systems demonstrate higher efficiency compared to Cas9, because it is oblivious to the location of the gene on the chromosome or on the plasmid [26,27,28,29]. In a recent study, it was shown that the *trans*-conjugative delivery system (Cas13a-based killing plasmid (CKP)) based on a *trans*-conjugative plasmid was effective in both in vitro and in vivo experiments. However, CRISPR-Cas13 acted on the transcriptome and had promiscuous RNA cleavage activity, resulting in the inability to eliminate plasmids carrying drug resistance genes. Finally, it had differential activation for different target sequences [30].

#### 2.1.2. CRISPR-Mediated Molecular Diagnostics

The ability to detect trace amounts of nucleic acids using the CRISPR-Cas system was used in molecular diagnostics since 2016 [31]. CRISPR-based diagnostic tools were intensively developed as an alternative to the PCR, since the latter requires special equipment, trained staff, which entails a time delay. A recent review by Liu, Frank X. et al. provided a detailed insight into the advances in CRISPR-based diagnostics with or without amplification for specific applications [32].

Molecular diagnostics, based on the activity of collateral cleavage of single-stranded DNA in Cas effectors, made it possible to effectively identify various viral strains: ZIKA virus, Dengue and human papilloma. Additionally, such diagnostics are considered for the parasite infection [33,34,35].

The features of human-to-human transmission and a high percentage of asymptomatic infections caused the severe acute respiratory syndrome coronavirus 2 (SARS-CoV-2) spread speedy and large-scale outbreaks worldwide, posing an extensive threat to health service [36]. A rapid and accurate application to detect SARS-CoV-2 is of great significance for COVID-19 control and elimination [37,38,39]. In recent years, the rapid and sensitive detection of SARS-CoV-2 was effectively realized by a collaborative system of CRISPR-Cas13a and graphene field effect transistors (GFET) biosensors [36]. Moreover, current research reports the ability of not only quick and effective detection of viral RNA in samples, but also to find out the differences between various options of the SARS-CoV-2 virus [37].

#### 2.1.3. Detection of Plant Viruses

The infection of significant crops with plant viruses causes economic losses by reducing crop quality and quantity globally. Therefore, it is required to develop multiple approaches to combat viral infection. At the same time, many antiviral strategies which were developed to date are limited to specific groups of viruses. Since single-stranded RNA (ssRNA) viruses are the most prevalent class of plant viruses, the application of CRISPR-Cas RNA-targeting systems is especially budding for crops. 

One of the Cas13 orthologues, CasRx, was found to be the most efficient and reliable variant for RNA virus interference in *Nicotiana benthamiana* [38]. Moreover, LshCas13a was employed to engineer interference with an RNA Turnip Mosaic Virus (TuMV) [39].

#### 2.1.4. CRISPR-Cas Systems in Detection of GMOs

In recent times, the biotechnology revolution and industrial transformation grew rapidly. Genome editing technology is actively used to improve crops; however, the release of such crops on the market or in the environment is of concern [40]. Thus, there was a need to detect genome-edited (GenEd) crops. RNA-targeted CRISPR/Cas systems can be used to detect genetically modified crops, namely crops with edited genes and single nucleotide polymorphisms (SNPs) [41]. Cas13a was used as a platform for the colorimetric detection of transgenic rice with platform sensitivity comparable to that of fluorescence methods [42]. Based on Cas13a, a platform for the simultaneous and sensitive detection of GM crops was created [43]. 

### 2.2. RNA Imaging and Tracking in Living Cells

The feature of a number of CRISPR-based RNA-targeting Cas nucleases to specifically bind and cleave target RNAs in the presence of guide RNA allows researchers to manipulate specific RNAs, involving the tracking of target RNAs and the editing of specific bases of RNAs [44,45]. To date, there are several methods which allow mapping RNA–protein interactions in the native cellular environment via CRISPR type VI effectors. The RPL (RNA proximity labeling), the CRUIS (CRISPR RNA-unified interaction system) and the CRISPR-assisted RNA–protein interaction detection (CARPID) methods are based on dead RNA-guided RNA targeting nuclease (dCas). It plays the role of a tracker to target specific RNA sequences, while the proximity labeling enzyme is fused to dCas to label surrounding RNA-binding proteins. Then, the results can be identified by mass spectrometry [46,47,48]. The advantage of all of these methods is that they require neither cross-linking nor any genomic manipulation of the genes encoding the RNA of interest.

### 2.3. Cancer Diagnosis and Therapy

Over the past few years, there was a huge interest around approaches to cancer treatment. The RNA-guided RNA-targeting CRISPR-Cas13 system has a high possibility for cancer diagnosis and therapy. In most CRISPR-Cas-based cancer therapy experiments, the LwCas13a was successfully used [44,45,46]. Nevertheless, among known Cas13 enzymes, RfxCas13d was the most effective ortholog discovered to date for RNA-targeting and found to have minimal off-target effects in mammalian cells [47,48], achieving knockdown of KRAS-G12D mRNA with up to 90% efficiency [49]. Furthermore, one more recent study offered a powerful tool for mRNA translation enhancement based on technology of catalytically inactive Type VI-D Cas13 enzyme (CasRx) and an integrated SINEB2 domain of uchl1 lncRNA. The CRISPR-dCasRx-SINEB2 system prevents the proliferation and migration of cancer cells both in vivo and in vitro. This effect was achieved by targeting the complex to antitumor proteins to increase their expression [50].

### 2.4. RNA-Editing and Regulation of Gene Expression

In the last 35 years, considerable progress was made in understanding the mechanisms of RNA-editing events and regulation of expression [51]. It is no wonder that there are various methods for RNA-targeting and regulating RNA expression. Such methods include antisense oligonucleotides (ASO), RNAi, CRISPRi and also the use of RNA-targeting CRISPR-Cas systems.

Tailor-made ASO constructs allow for knockdown and demonstrate their effectiveness in the treatment of diseases caused by dysregulation of splicing events [52]. However, this technology is quite expensive and also has a transient life cycle, which may require a lifetime introduction of ASO. RNAi technologies were also demonstrated to be highly efficient, but prone to significant off-target effects. In addition, RNAi reagents can interfere with the endogenous miRNA pathways and silence untargeted transcripts [53]. The CRISPRi technology uses Cas effectors with killed nuclease activity, fused with transcription activators or repressors. It allows to regulate RNA at the DNA level and does not allow modulate transcripts resulting from alternative splicing events, thereby limiting the use of CRISPRi. As a result, there is still a need for reliable and specific RNA-targeting tools.

#### 2.4.1. RNA Knockdown and Gene Silencing

Due to their programmability and ability to cleave target RNAs through RNase activity, RNA-targeting CRISPR effector proteins can be used to suppress the expression of target genes. It was demonstrated that RfxCas13d silences expression of genes associated with neurodegenerative disorders (amyotrophic lateral sclerosis and Huntington’s disease) [54]. The StCsm complex also demonstrated efficient knockdown of the EGFP gene in Zebrafish [55]. Furthermore, the efficiency of RNA knockdown was demonstrated in O. sativa protoplasts [44].

Despite the above successes in applying CRISPR effectors as a platform for suppressing gene expression, there were some limitations. It is obligatory for an effector protein to be constantly expressed in cells, so it can always interact with the target mRNA to maintain the effect. In addition, the rapid degradation of crRNA may become a problem which can be overcome by introducing chemical modifications to crRNA [56]. Furthermore, such proteins can be recognized as foreign by the immune system, thereby causing immunogenic effects. Finally, long-term studies are required to determine the effects which may result from sustained expression of Cas effectors.

#### 2.4.2. Splicing Alteration

One approach to redirecting splicing is to target critical regions beside intron–exon junctions, which can block the recognition of splicing factors and control the production of mature mRNA. Such targeting can be achieved using RNA targeting Cas effectors, which was already shown for dCasRx [45]. As one of the most compact single effector Cas enzymes, dCasRx was successfully packaged into an adeno-associated virus and delivered to a neuronal model of frontotemporal dementia, where it demonstrated alteration of tau isoform ratios, through manipulation of alternative splicing and inducing exon exclusion [45].

#### 2.4.3. Enhancement of Protein Translation

The ability to regulate gene expression at the translational level can be of considerable interest as a tool for studying fundamental biological phenomena, as well as a new therapeutic strategy. Although there are several effective strategies for manipulating gene expression at the genomic or transcriptional level, only a few of them are available for modulating translation.

RNA-targeting CRISPR-Cas systems make it possible to target mRNA and can be used to regulate expression at the post-transcriptional level, which was demonstrated in a cell-free expression system. In a recent study, a significant reduction in protein expression was achieved through sgRNA targeting to the start codon of the sfGFP (superfolder green fluorescent protein) mRNA, thereby precluding ribosome binding [57]. Furthermore, combining RNA-targeting dCas systems with various regulatory elements, it can be possible to achieve not only a decrease in the level of expression, but also a significant increase in it. It was shown that the complex dCas13Rx-sgRNA with the SINEB2 element can increase translation of its target mRNA. The advantage of such RNA-targeting CRISPR-Cas systems is that they can influence the expression of target proteins without inducing stable genetic changes in cells.

#### 2.4.4. RNA-Targeting CRISPR-Cas Systems for RNA Editing

RNA editing can be performed through engineering RNA-targeting Cas effectors in the way that they retain the ability to bind target RNA in a controlled manner, additionally removing nuclease activity and fusing it with ADAR (adenosine deaminase acting on RNA). 

Cas-based RNA base editors can be applied for mutation correction in the treatment of various diseases. The dCas13X.1-based adenosine base editor was shown to exhibit high efficiency of A > G conversion and low frequency of off-target edits on mouse models of deafness [58]. 

RNA editing tools provide reversible editing of mutant transcripts. On the one hand, it reduces the potential risk of permanent genetic changes, similar to DNA editing tools. On the other hand, RNA-editing therapy implies its continuous insertion.

Since the discovery of RNA-targeting nucleases, much effort has been made to apply them as an RNA-target editing tool in eukaryotic cells. Nevertheless, many recent reports highly suggest that RNA-targeting CRISPR-Cas systems behave unpredictably in eukaryotic cells. RNA-targeting CRISPR effector proteins are toxic to cells due to their side RNase activity. Therefore, many developed systems for the regulation of gene expression have a critical drawback for in vivo applications. However, Cas13 variants were already generated that retained high target RNA cleavage activity but showed minimal collateral cleavage activity [59]. Collateral cleavage activity in eukaryotic cells by RNA-targeting CRISPR-Cas does create new possibilities in biosensing applications of the versatile CRISPR-Cas RNA targeting systems. 

## 3. CRISPR Toolbox

### 3.1. Type VI CRISPR-Cas

The CRISPR-Cas13 system consists of single RNA-guided Cas13 effector nucleases that solely target single-stranded RNA (ssRNA) in a programmable way without altering the DNA [7]. In recent times, based on phylogeny, type VI nucleases is ranked into VI-A (Cas13a ~ 1250 a.a.), VI-B (Cas13b ~ 1150 a.a.), VI-C (Cas13c), VI-D (Cas13d ~ 930 a.a.), VI-E to I (Cas13e-Cas13i), and the latest CRISPR-Cas13 system such as Cas13X (~775 a.a.), Cas13Y and Cas13bt (~775–804 a.a.) [60,61,62,63]. Cas13 effectors have a bi-lobed structure: NUC (nuclease) lobe and a REC (recognition) lobe. The NUC lobe consists of two functionally linked higher eukaryotes and prokaryotes nucleotide-binding domains—HEPN-1 and HEPN-2 [64]. The REC lobe encompasses the N-terminal domain (NTD) and the Helical-1 domain [65]. To cleave RNA molecules Cas13 nuclease requires a guide from a matured crRNA [33,66]. Despite the fact that CRISPR type VI does not possess endonuclease activity for processing pre-crRNA into matured crRNA, Cas13 performs both crRNA biogenesis and RNA cleavage employing two chemically and mechanistically distinct mechanisms for both tasks [25]. It was revealed that Cas13 cleaves 2-5 nucleotides upstream of the target pre-crRNA to produce a 60–66 nucleotide-long matured crRNA [67]. The resulting matured crRNA contains a single spacer sequence (20–30 nucleotide long) [68] (Table 1). 

### 3.2. Type V CRISPR-Cas

Type V CRISPR-Cas systems are remarkable for a single RNA-guided RuvC domain-containing Cas12 effector. Cas12g is a recently identified type V RNA-guided endonuclease. Following activation by target molecular RNA, Cas12g shows collateral RNase and single-strand DNase activities, similar to the collateral activity of Cas12a and Cas13 nucleases. Cas12g effector displays a canonical bilobed architecture containing a recognition lobe (REC) and a nuclease lobe (NUC) [10,69,70,71]. To achieve a successful targeting of ssRNA, Cas12g applies two guide RNAs, a tracrRNA and a crRNA, without requiring protospacer adjacent motif (PAM) sequence [16]. Additionally, Cas12g nucleases have a compact size, compared to type VI nucleases (only 767 a.a.) [17].

### 3.3. Type III CRISPR-Cas 

Type III CRISPR–Cas is believed to be the oldest member of the CRISPR–Cas family [72]. Recently, six different type III subtypes were identified: III-A to III-F [73]. The effector complexes consist of multiple subunits (Cmr1, Cmr3, Cmr4, Cmr5 and Cmr6 in types III-B and III-C, and Csm2, Csm3, Csm4 and Csm5 in types III-A and III-D), with key subunit Cas10 (Cmr2 in types III-B and III-C, and Csm1 in types III-A and III-D) being the largest component [23,74,75]. Although most of the type III-associated proteins are not characterized in detail, type III CRISPR–Cas seems to be a skilly regulated system, displaying properties such as signal amplification, self-regulation and controlling of signaling molecule concentrations [23]. CRISPR-Csm uses an RNA-guided mechanism to cleave target RNA molecules while not demonstrating trans-cleavage activity. Like other RNA-targeting CRISPR-Cas systems, Csm does not have PAM specificity for target site selection [76]. The multiprotein Csm complex consists of five different proteins: Cas10, or Csm1, Csm2-Csm5 and relies on an additional protein, Cas6, for processing the precursor crRNA. The crRNA is located at the core of the Csm complex and has eight nucleotides at its 5′ end originating from CRISPR repeat and 30–45 nucleotides in guide sequence, derived from one of CRISPR array spacers [77,78,79,80,81]. The single-stranded DNase activity belongs to the Csm1 subunit, and each Csm3 subunit has ribonuclease (RNase) activity [77,82,83,84]. Due to their structural complexity, multisubunit Cas effectors are used as a genome editing tool in eukaryotic cells on rare occasions [85,86]. However, the type III-A Csm complex from S. thermophilus was recently shown to be a powerful tool for eukaryotic RNA knockdown. Both nuclear noncoding RNAs and cytoplasmic mRNAs were eliminated with high efficiency (90–99%) and specificity (~10 times less off-target effects than Cas13) [76].

### 3.4. Type II CRISPR-Cas

The type II CRISPR–Cas system belongs to class 2 and is divided into three subtypes (II-A, II-B and II-C). In addition to CRISPR array and Cas proteins, this type encodes additional trans-activating RNA (tracrRNA), which mediates interaction between crRNAs and Cas9 [87]. Type II CRISPR-Cas systems were considered mainly as RNA-guided double-stranded DNA endonucleases, while the ability of various Cas9 homologous to cleave RNA substrates was less affected. Cas9 in complex with guide RNA binds to DNA only if it contains the PAM sequence, which allows the immune system to distinguish between foreign DNA. Based on the fact that PAMs recruit RNP complexes to potential target sites and trigger nuclease domain activation, ssDNA cleavage experiments were performed using PAMmer—PAM-presenting oligonucleotide. The same logic was used to retarget Cas9 to bind and activate the ssRNA cleavage process [20]. The study showed that only the deoxyribonucleotide PAMmer lead to cleavage, while the ribonucleotide did not. It was later shown that not all Cas9s required PAMmer for RNA cleavage. Comparison of the behavior of various Cas9 homologous in the context of directed cleavage of single-stranded RNA showed that SpyCas9 II-A and FnoCas9 II-B require PAMmer to cut RNA, while CjeCas9 II-C and SauCas9 catalyze directed and PAM-independent cleavage of single-stranded RNA by sgRNA [18]. Additionally, it was shown that RNA binding was crRNA- and tracrRNA-dependent, and that target RNA cleavage required the CjeCas9 HNH domain [87]. Thus, some Cas9 homologous can be used to perform PAM-independent RNA targeting, which requires sgRNA complementary to the target RNA and, for some Cas9, PAMmer. However, RNA-cleavage by type II CRISPR-Cas systems is slower than DNA-cleavage and kinetically less robust than that of Cas13 [18]. The influence of target RNA secondary structures on the cutting efficiency can also be attributed to the limitations of the application (Table 1).

**Table 1 ijms-24-06894-t001:** Main characteristics of RNA-targeting CRISPR systems.

Effector	Characterizations
Cas13	Two HEPN nuclease domains
	Collateral RNA cleavage No PAM requirements Suffer from severe cytotoxic effects due to *trans*-cleavage activity Processes own crRNA Single-subunit effector Small enough to fit a single AAV vector with a crRNA expression cassette Spacer length ~ 25 nt Adapted to serve numerous applications Extensively characterized
Cas12g	Collateral RNase and ssDNase activities
	Single-subunit effector No PAM requirements Compact size Applies two guide RNAs, a tracrRNA and a crRNA Insufficiently explored
Csm	Multiprotein complex No collateral activity
	No PAM requirements
	Powerful tool for eukaryotic RNA knockdown Fewer off-targets than Cas13 RNase and DNase activity crRNA lies at the core of the complex Spacers crRNAs range from ~30 to 45 nt Extensively characterized
Cas9	Contains two nuclease domains (HNH and RuvC) Collateral activity not identified PAM is required
	Single-subunit effector Spacer length is 20 nt Low off-target product Target dsDNA and RNA Relies on RNase III to process its crRNA Extensively characterized

## 4. Conclusions

The past 10 years testified great progress in the understanding and manipulation of CRISPR-Cas systems, especially in the field of genome editing and RNA editing. The innovation of CRISPR-Cas changed the conception of biosensing systems and also allowed the CRISPR effectors to be used in various applications; for example, genomic editing, effective virus diagnostic tools, biomarkers, transcription regulations. In just a few years, RNA-targeting CRISPR-based biosensors evolved from a hypothetical nucleic acid sensing tool to an opportunity for diagnostic technology for the rapid, reasonable and ultrasensitive sensing of biomarkers. In addition, Cas13-based biosensors are not expensive, sensitive, selective and easy to apply for the detection of DNA, RNA and proteins in liquid biopsy samples. Technologies based on CRISPR-RNA targeting systems may become potent new tools for controlling protein expression in biological research and biopharmaceuticals production. Finally, CRISPR-Cas13d and engineered variants such as CasRx enable nucleic acid and transcriptome engineering for future development.

Each CRISPR-based RNA targeting method has complex and sometimes seemingly insurmountable challenges. Unfortunately, there are still some difficulties in applying well-characterized effector Cas13a for genome editing in mammalian cells. The rapidly growing interest in type III nucleases suggests that, in the future, we will have a versatile RNA-targeting tool, although it is currently not often used in practice. A satisfactory characterization of an effector in vitro cannot guarantee the same efficiency in living cells. As for Cas12g, their usage is limited with the lack of enough characterization data. Nuclease Cas9 does not have *trans*-cleavage activity and it is mainly characterized in the frame of DNA targeting applications. Additionally, not all orthologs of Cas9 are PAM-independent. 

Despite all aforementioned problems, the CRISPR-Cas RNA-targeting systems already demonstrated their efficiency for a range of applications. In recent works, scientists from all over the world conducted much research on currently available effectors. Due to this, the structures, characterization of specificity and *trans*-cleavage activity guide RNA requirements and kinetic characteristics became available for us. All received data lay the groundwork for scientists to expand the scope of applications of previously obtained systems at the expense of an efficient design and directed evolution to offset any shortcomings existing in RNA-targeting systems. Finally, identification and characterization of new CRISPR-Cas RNA-targeting systems from metagenomic data may be considered to establish and consummate their employment. 

Deciphering the full range of RNA-targeting CRISPR-Cas systems, with its many uncharacterized proteins and functionalities, will provide scientists all over the world with experimental provocations for years to come as well as potential for new incredible applications. In the end, technologies based on RNA-targeting CRISPR-Cas systems will facilitate progress in scientific and treatment applications, paving the pathway for the creation of new technologies for research and biotechnology.

## Figures and Tables

**Figure 1 ijms-24-06894-f001:**
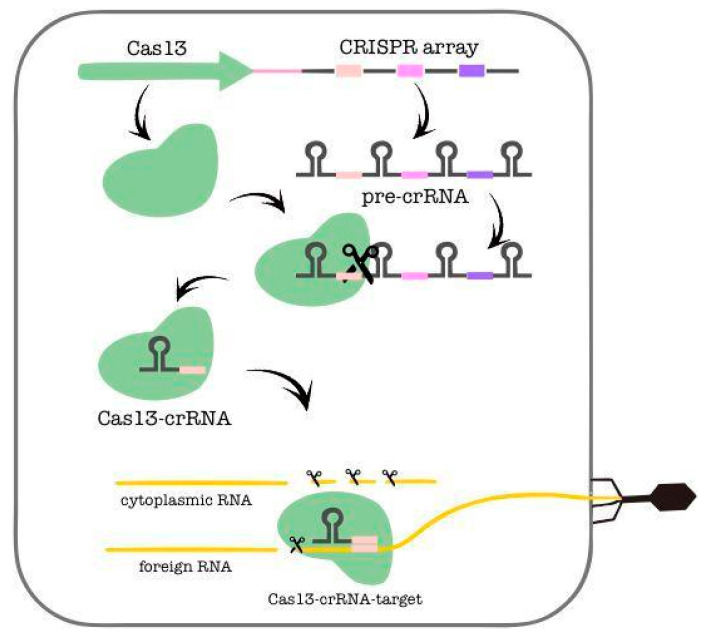
Mechanism of the CRISPR–Cas13 (type VI) system.

**Figure 2 ijms-24-06894-f002:**
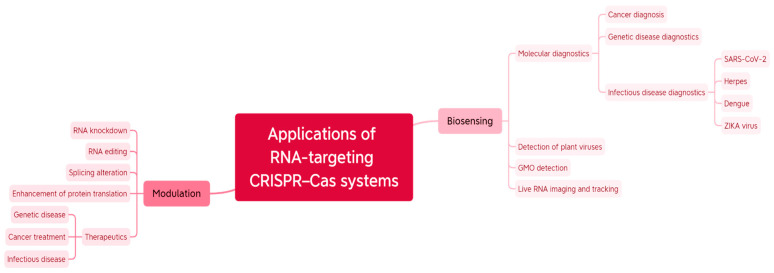
RNA-targeting CRISPR systems applications.

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
