# Peer review of "RNA-Dependent RNA Targeting by CRISPR-Cas Systems: Characterizations and Applications"

_ijms, 2023, doi:10.3390/ijms24086894_

Round 1

Reviewer 1 Report

This article nicely and structurally summarized different classes of CRISPR-Cas systems such as class II, III, V, and VI that have been reported to have activity with RNA molecules. In addition, the article also provides a summary of CRISPR-Cas systems for various RNA applications that are covered in various fields including, GMO detection in crops, disease detection in humans and crops, treatment, biosensors, and more. This article helps the reader to navigate which CRISPR-Cas system is suitable for which application. There is no major concern for me. Listed below are minor concerns.

-       Line 51 “Recognition of RNA molecules circumvents hindrance by DNA modifications. What type of hindrance did the author discuss here? Examples and citations.

-       Line 51. The two rationales that the authors try to convince that RNA editing is better than DNA editing are difficult to understand. Please elaborate more on this.

-       Shall Section 3 come before applications?

-       Line 30 guide RNA (sgRNA) to single guide RNA (sgRNA)

-       Line 45 Citations 8-12, I don’t think these citations are appropriate. They should be of original research articles that showed Cas systems that can indeed correct or recover the function of genes that are impossible to replace by gene therapy.

-       Citation should be in numerical order; however, they seem to jump around a bit. For example, citations on line 53 and line 165

-       Line 132. It would be nice to have a report from a study to show that it can detect GMOs and SNP in crops

-       Line 206 needs a citation

-    Line 156 citation 44 is an incorrect citation

Reviewer 2 Report

The understanding and application of RNA-targeting CRISPR-Cas systems are interesting topics in this field. In this review, Gunitseva et al. attempt to summarize the current progress on the potential applications of the versatile RNA-targeting systems. In general, this reviewer thinks the authors covered most of the key CRISPR subtypes that were considered to have RNase activity, for instance, type II (Cas9), type III, type V, and type VI systems. However, at least one figure is suggested to be added in the revised version. This reviewer thinks it is a good idea to include the mechanism of some of these CRISPR systems in the suggested figure. Meanwhile, it is better to optimize the current layout of Figure 1 in the manuscript. In addition, more information was needed to discuss CRISPR-based RNA imaging and tracking in living cells.

The manuscript needs mederate editing and particular attention to English grammar and sentence structure. Just two examples are listed:

Lines 86-87, add a comma after “study”

Line 348, Clerical error, it should be “Technologies~~~”
